# Unsupervised inter-domain transformation for virtually stained high-resolution mid-infrared photoacoustic microscopy using explainable deep learning

Eunwoo Park [1,2,10], Sampa Misra[1,2,10], Dong Gyu Hwang [2,3,10], Chiho Yoon [2,4], Joongho Ahn [2,4,5], Donggyu Kim[1,2], Jinah Jang [1,2,3,6,7,8] ✉ & Chulhong Kim [1,2,4,5,6,7,9] ✉

Mid-infrared photoacoustic microscopy can capture biochemical information without staining. However, the long mid-infrared optical wavelengths make the spatial resolution of photoacoustic microscopy significantly poorer than that of conventional confocal fluorescence microscopy. Here, we demonstrate an explainable deep learning-based unsupervised inter-domain transformation of low-resolution unlabeled mid-infrared photoacoustic microscopy images into confocal-like virtually fluorescence-stained high-resolution images. The explainable deep learning-based framework is proposed for this transformation, wherein an unsupervised generative adversarial network is primarily employed and then a saliency constraint is added for better explainability. We validate the performance of explainable deep learning-based mid-infrared photoacoustic microscopy by identifying cell nuclei and filamentous actins in cultured human cardiac fibroblasts and matching them with the corresponding CFM images. The XDL ensures similar saliency between the two domains, making the transformation process more stable and more reliable than existing networks. Our XDL-MIR-PAM enables label-free high-resolution duplexed cellular imaging, which can significantly benefit many research avenues in cell biology.

Confocal fluorescence microscopy (CFM) is the current gold standard for high-resolution (HR) imaging techniques in the life sciences and biomedicine[1,2]. Based on the excitation of fluorophores, various fluorescence (FL) dyes distinctively specify biomolecules of interest in a reliable and flexible manner. Using multiple excitation and emission channels enables multiplexed imaging, offering cellular and sub-cellular biological insights[2–5]. However, FL staining is time-consuming and causes photobleaching and phototoxicity, challenging stable

[1]Department of Convergence IT Engineering, Pohang University of Science and Technology (POSTECH), Pohang, Republic of Korea. [2]Medical Device Innovation Center, Pohang University of Science and Technology (POSTECH), Pohang, Republic of Korea. [3]Center for 3D Organ Printing and Stem Cells, Pohang University of Science and Technology (POSTECH), Pohang, Republic of Korea. [4]Department of Electrical Engineering, Pohang University of Science and Technology (POSTECH), Pohang, Republic of Korea. [5]Opticho Inc, Pohang, Republic of Korea. [6]Department of Mechanical Engineering, Pohang University of Science and Technology (POSTECH), Pohang, Republic of Korea. [7]Department of Medical Science and Engineering, Pohang University of Science and Technology (POSTECH), Pohang, Republic of Korea. [8]Institute for Convergence Research and Education in Advanced Technology, Yonsei University, Seoul, Republic of Korea. [9]Graduate School of Artificial Intelligence, Pohang University of Science and Technology (POSTECH), Pohang, Republic of Korea. [10]These authors contributed equally: Eunwoo Park, Sampa Misra, Dong Gyu Hwang. ✉e-mail: jinahjang@postech.ac.kr; chulhong@postech.edu

cellular imaging. The unstable readout of the FL signal, varying with the imaging conditions or the status of the labeled sample, makes it difficult to quantitatively analyze the FL intensity.

An alternative to CFM, photoacoustic microscopy (PAM) is a promising biomedical imaging technology based on the light absorption of chromophores[6–8]. Due to the strong light absorption in endogenous tissue pigments, certain biomolecules can be spectrally distinguished without labeling[9–15]. The majority of chemical specificities are observed as vibrational transitions in the infrared bands. Chemical molecular bonds and functional group specificities are identified by vibrational overtones and combinations of stretching and bending modes. Particularly, signatures in the mid-infrared (MIR) region can further distinguish chemical compositions with distinctive spectral regions[16]. By detecting photothermal effects in the functional group and fingerprint spectral regions, MIR-PAM can provide bond-selective imaging based on vibrational absorption contrast. Previous MIR-PAM studies demonstrated label-free imaging of lipids, proteins, and carbohydrates, with high sensitivity. Multi-spectral MIR-PAM using fresh biological samples has been demonstrated in applications ranging from metabolic imaging at the cellular level[17] to histological imaging at the tissue level[18,19]. However, the optical diffraction of long wavelengths limits the spatial resolution of MIR-PAM, posing challenges to obtaining cellular-level HR images. Although ultraviolet (UV) localization has improved the lateral resolution of MIR-PAM[18], the requisite complex hardware configuration, as well as UV light-induced photodamage, degrades the stability of the imaging system. To ensure reliable cellular imaging with MIR-PAM, high-resolution and subcellular feature identification should be accompanied.

Rapid advances in deep learning (DL)-approaches have revolutionized image processing[20–26]. To highlight crucial information or reveal previously inaccessible data, DL-based image transformation translates one image domain into another[27,28]. U-Net, one of the most well-known convolution neural networks (CNNs), has excelled in resolution enhancement[29] and virtual staining[30]. In addition, a generative adversarial network (GAN) employs a perceptual-level loss function to generate more accurate results[30–32]. However, these familiar supervised DL methods heavily rely on large amounts of high-quality training data and registered annotations. The innovation of cycle-consistent GAN (CycleGAN) offered a breakthrough by enabling unsupervised training without the requirement of strictly matched image pairs[33]. CycleGAN presents a practical strategy for cases where only unpaired images are available, and it is especially useful in style transfer[34,35] and biomedical imaging studies[36,37]. Recently, there has been a growing interest in the explainability of DL models. Although the sophisticated mathematics underlying DL training algorithms is conceptually understandable, the algorithms' architectures are more of a black box model. Achieving an explainable DL model would help developers troubleshoot problems and better explain to clients why a certain outcome is predicted by the model. In particular, explainable DL (XDL) for unsupervised training facilitates effective feedback by visualizing the features that contribute most to the outcome[38].

Here, we present XDL-based MIR-PAM (XDL-MIR-PAM), a system that achieves HR duplexed PA imaging at the cellular level without any FL staining. A basic overview of XDL-MIR-PAM is shown in Fig. 1. Cellular-level images are obtained by two independent optical imaging modalities: MIR-PAM provides low-resolution (LR) protein-selective imaging in unlabeled cells using a monochromatic wavelength, whereas CFM provides HR multiplexed imaging in immunofluorescent stained cells using multiple wavelengths. Figure 1a shows the workflow for the unsupervised inter-domain transformation (UIDT). The UIDT process has two components: (1) an image resolution enhancement network (IREN) and (2) a virtual FL staining network (VFSN). Even with unpaired image sets, these networks respectively transform LR images into HR ones and transform unlabeled intensity images into virtually stained (VS) images. In addition, the UIDT networks, based on the CycleGAN model, transform images in the other domain, and saliency image similarity metrics are employed in the networks to achieve explainability for the DL models (Fig. 1b and Supplementary Fig. 1). To correct the mapping direction and prevent image content distortions, the explainable CycleGAN imposes a saliency constraint in addition to the cycle-consistency loss functions. The saliency mask maintains a high degree of similarity in each transformation. Consequently, by integrating both domain advantages with superior reliability, XDL-MIR-PAM has great potential as a groundbreaking imaging technology in biological research.

## Results

### Label-free protein-selective MIR-PAM

To demonstrate protein-selective MIR-PAM, human cardiac fibroblasts (HCFs), stromal cells in the cardiac tissue, were photoacoustically imaged. We employed Fourier-transform infrared (FTIR) spectroscopy to find chemical bonds in the HCFs. Figure 2a shows the optical absorbance of HCF calculated by the averaged FTIR spectra. The main absorption peaks of proteins appear at $1652\,cm^{-1}$ for the amide I bond and $1547\,cm^{-1}$ for the amide II bond. The peak at $1083\,cm^{-1}$ arises from the absorption of phosphorylated molecules ($PO_2^-$ symmetric stretching) and C−O bonds evidencing nucleic acids or phospholipids. Accordingly, a tunable quantum cascade laser (QCL) was used as a light source in the MIR-PAM, and the center wavenumber was set to $1667\,cm^{-1}$ (6.00 μm) for label-free protein-selective imaging. The MIR-PAM imaging system is shown in Fig. 2b (see the details in "Methods"). The laser beam was efficiently delivered through a dry nitrogen chamber, mitigating light attenuation from water vapor, and cells were seeded on a zinc selenide (ZnSe) plate, which was highly transparent over the wide IR band. The reflective objectives focused the MIR light on the target cells, and the generated PA signals were detected by an ultrasound transducer (UST) with a central frequency of 30 MHz. Because scanning was driven by motorized XY stages, whose motion might dislodge the cells, the ZnSe plate was coated with fibronectin for cell adhesion. Label-free MIR-PAM images of HCFs on days 1 and 7 are shown in Fig. 2c. The HCFs are visualized based on protein selectivity without FL staining in the cell nucleus and filamentous actin (F-actin). 16 images were used each on days 1 and 7. During the HCF growth, the averaged PA signal amplitude increases by about 1.40 times, implying that the amount of protein increases (Fig. 2d). In the enlarged view shown in Fig. 2e, both images show high PA amplitudes in the cell nuclei, one of which is indicated by white arrows. On day 7, more F-actin is expressed, and HCF elongation is observed (green arrow). Therefore, the overall cell confluency increased from $51.2 \pm 2.1\%$ to $75.7 \pm 3.9\%$ (Fig. 2f). By using label-free MIR-PAM for protein-selective imaging of HCF, the cell expression and growth can be quantitatively inferred. However, the image resolution is not sufficient to distinguish detailed structures at the cellular level. The developed MIR-PAM has a lateral resolution of about 6.6 μm with an imaging depth of about 60.7 μm (Supplementary Note 1 and Supplementary Fig. 2).

### XDL-based image resolution enhancement network (XDL-IREN)

Introduced in Fig. 1a, the IREN, a DL network, enhances lateral resolutions with transformation between imaging modalities. The IREN aims to predict HR-MIR-PAM images. MIR-PAM and CFM images are utilized as LR input and HR target domains, respectively (Fig. 3a). These image collections in both domains do not require pre-aligned image pairs. After independently acquiring images in the source and target domains, we randomly divided the whole-slide images from each domain into ~ 900 tiles of 256 × 256 pixels to create the training dataset. Such a minimal input size can speed up the training process and decrease memory requirements. No paired images were included in the two training sets. Then, we trained the IREN on this dataset to

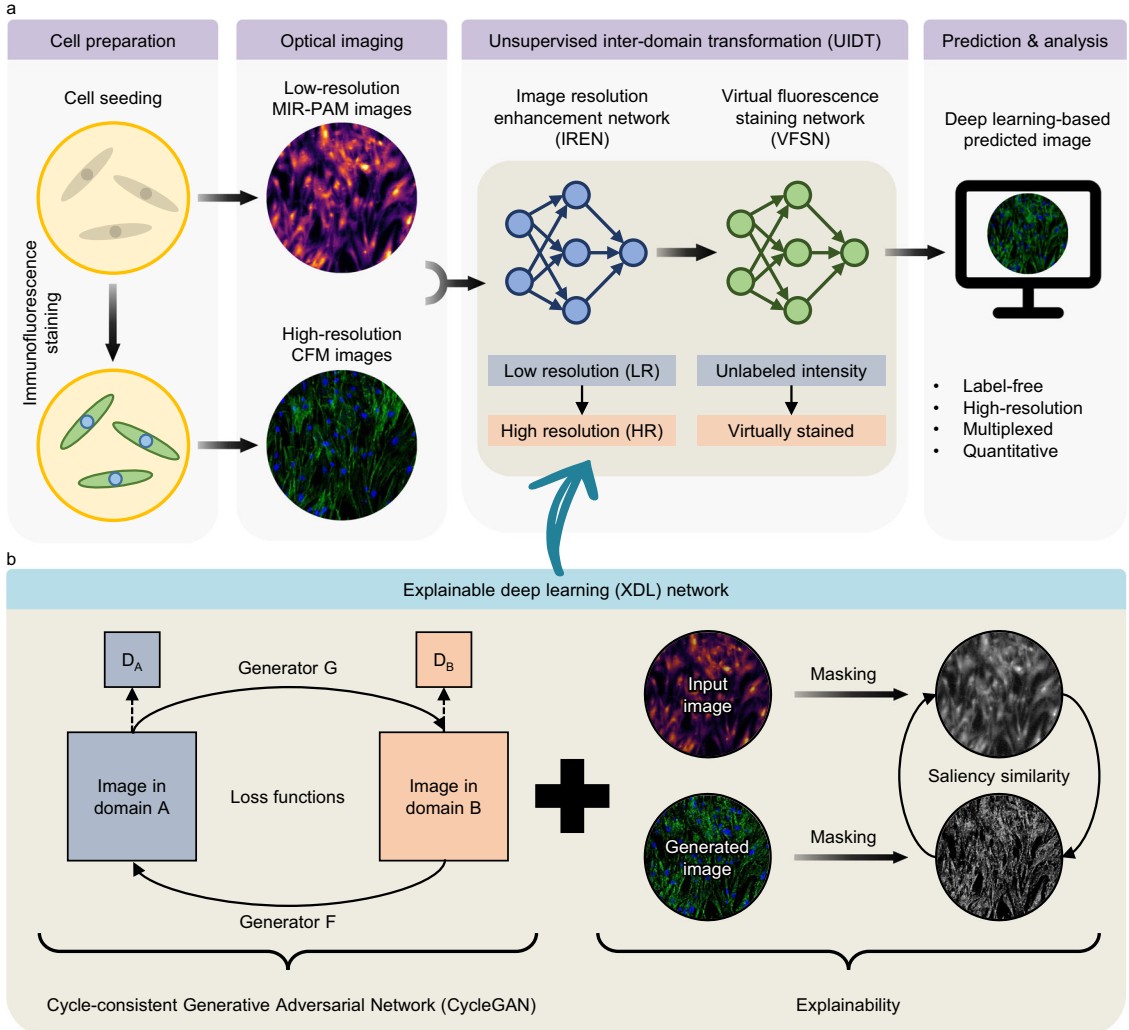

**Fig. 1 | Overview of the XDL-MIR-PAM. a** Workflow for UIDT in MIR-PAM images. Low-resolution MIR-PAM images and high-resolution CFM images of cultured cells are inputs of UIDT. The two-step UIDT produces high-resolution and virtually fluorescence-stained images of label-free cells. **b** The network configuration for the XDL. By adjusting the saliency masks between the input and generated images, the XDL model adopts a saliency similarity in loss functions of the existing network to achieve explainability. MIR-PAM, mid-infrared photoacoustic microscopy, CFM, confocal fluorescence microscopy, and $D_A$ and $D_B$ denote discriminators of each domain.

learn the transformation from the LR domain to the HR one. In the CycleGAN loss restrictions, we added a saliency constraint as well as a structural similarity index (SSIM) to increase the explainability of the model (Fig. 3b). Saliency loss continuously tracks saliency masks for both image domains to address unexpected errors that inevitably occur during the training process (Supplementary Fig. 3). In addition, GradCAM[39] technique is used to further explain the inner behavior of the model during the domain transformation from LR to HR (Supplementary Fig. 4). GradCAM heatmaps show the model's attention in each transformer layer of the XDL-based generator. GradCAM captures the morphological and textural features in HCF that contribute to the transformation, and the key features become apparent as the layer progresses. Notably, the cell nucleus is consistently highlighted and structurally distinguished from F-actin.

This XDL-IREN has greater stability and dependability than the traditional DL-based one because the saliency similarity in the IREN makes sure that the extracted saliency mask of the input LR image stays consistent when transferred to the HR domain. Without this restriction, the output HR image would be deformed and lose the semantic content of the LR images. Figure 3c tabulates the DL performance comparison resulting from the loss restrictions. Compared

to the scores for the DL-IREN, for the XDL-IREN, the calculated Frechet inception distance (FID) and kernel inception distance (KID) scores of the transformed images are decreased by 71.0 and 9.5, respectively. The XDL-IREN with both constraints shows the best performance. In terms of image fidelity, a low FID score indicates that the feature distance of the multivariate distribution is close, and a low KID score indicates that the maximum mean discrepancy of features between the synthesized and real images is small (see the details in "Methods"). Figure 3d compares images processed by the proposed IRENs with conventional DL and XDL. Here, the conventional DL network was assigned to the original CycleGAN model, while the model with the addition of SSIM and saliency loss restrictions was assigned to the XDL network. The IREN can transform the LR-MIR-PAM images into the HR-MIR-PAM ones while preserving the original structures of the given CFM images (the ground truth). Cell nuclei and F-actins appear in oval and linear structures, respectively. In particular, the XDL-IREN improves the LR-MIR-PAM images over the DL-IREN, which are more comparable to the HR images. As shown in the magnified view, the XDL-IREN-based HR-MIR-PAM images reveal more features of F-actins that are not shown by the DL-IREN (yellow arrows), and they preserve the contents of the cell nucleus (green arrows). Thus,

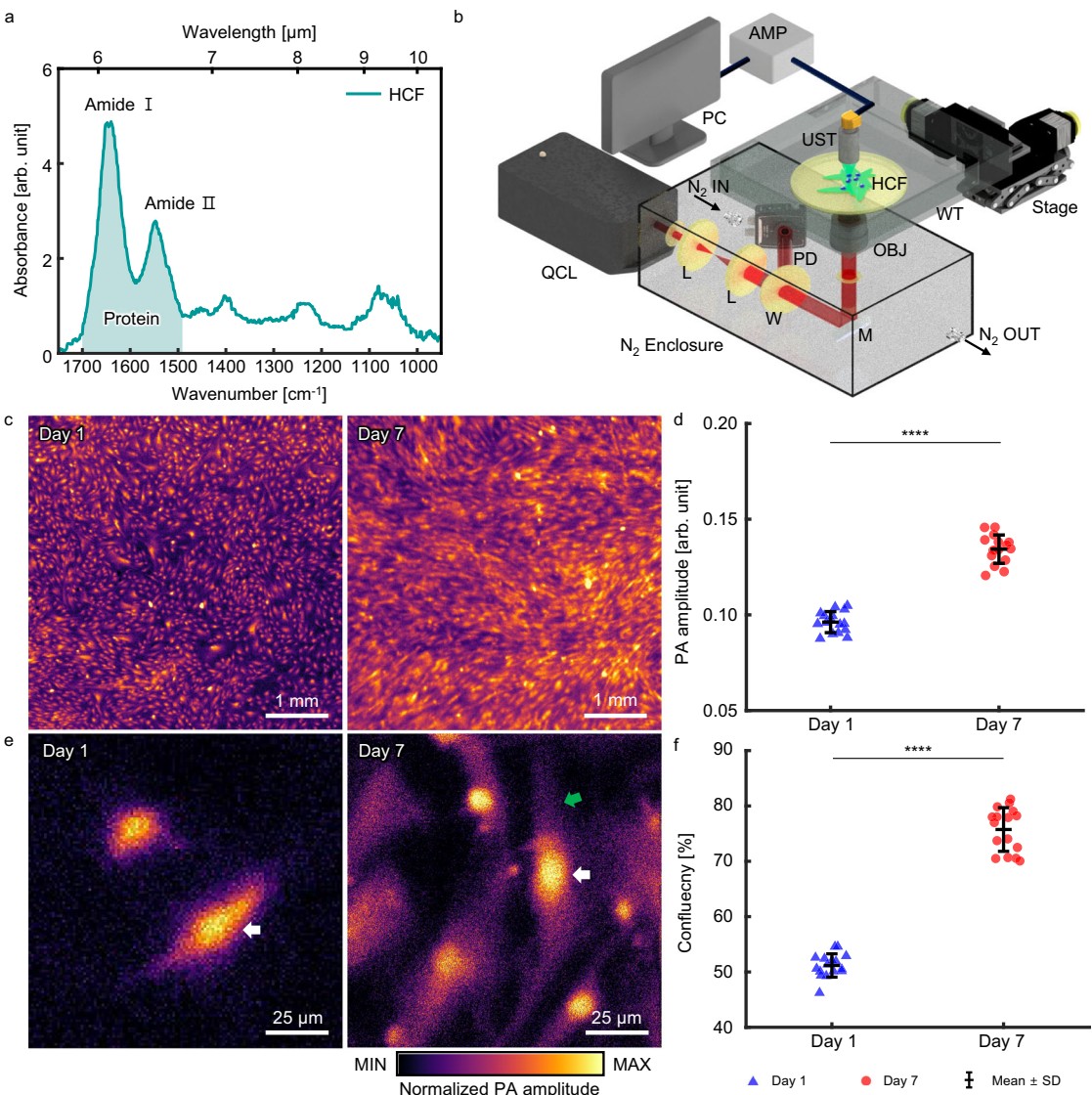

**Fig. 2 | Label-free MIR-PAM of HCFs. a** Fourier-transform infrared spectrum of HCFs. **b** Schematic diagram of MIR-PAM system. **c** Label-free MIR-PAM images of HCFs at days 1 and 7 (pseudo-colored). **d** Averaged PA signal amplitudes ($n = 16$, mean ± SD). Significance by unpaired two-tailed $t$ test: ****, $p = 1.3 \times 10^{-16}$. **e** Magnified view images of Fig. 2c. **f**, Cell confluency at day 1 and day 7 of growth. ($n = 16$, mean ± SD). Significance by unpaired two-tailed $t$ test: ****, $p = 4.9 \times 10^{-20}$. QCL, quantum cascade laser; L, Lens; W, window; M, Mirror; OBJ, Objective lens; WT, water tank; HCF, human cardiac fibroblasts; UST, Ultrasonic transducer; and AMP, Amplifier. Source data are provided as a Source Data file.

the XDL-IREN-generated HR-MIR-PAM images can capture detailed structures of HCF (1–2 μm) beyond the resolution of LR-MIR-PAM (6–7 μm) (Supplementary Fig. 5).

**XDL-based virtual fluorescence staining network (XDL-VFSN)**
Following the resolution enhancement in Figs. 3a, 4a demonstrates the operating sequence of the XDL-based VFSN for CFM images of HCFs. We used HR-CFM images as the training dataset for virtual staining. Grayscale (i.e., unlabeled) and colored CFM images are adopted for the input and ground truth domains, respectively. The VFSN aims to transform unlabeled CFM intensity images into virtually FL-stained ones with biological specificity. The training procedure of the VFSN is similar to that of the IREN. Using the unpaired training dataset, each domain contains ~900 image tiles. The unlabeled CFM images (i.e., input), two types of VFSN predicted images (one with conventional DL and the other with the XDL), and the corresponding ground truths are shown in Fig. 4b. Here, we employ saliency similarity to increase the model's explainability and compare the results between the CycleGAN

(conventional DL) and explainable CycleGAN (XDL) in the VFSN. In typical FL staining of HCFs, cell nuclei, and F-actins are visualized by blue-FL Hoechst and green-FL fluorescein (FITC) stains, respectively. While the DL-VFSN shows non-specific staining errors (yellow arrows), the XDL-VFSN achieves a more realistic label distribution and better global effects. Without prior annotation or segmentation, chromatic channel separation of cell nuclei and F-actins is feasible. We further quantified the performances of the VFSN by calculating the SSIM, peak signal-to-noise ratio (PSNR), Pearson's correlation coefficient (PCC), FID, and KID for 98 test image tiles (Fig. 4c). All scoring metrics were significantly improved by the XDL-VFSN, demonstrating its robustness. This VFSN enables highly reliable labeling in the unlabeled CFM images domain and is envisaged to be linked with IREN.

**Framework configurations of XDL-based unsupervised inter-domain transformation (XDL-UIDT)**
By integrating two prebuilt DL networks (i.e., the IREN and VFSN), we configured the frameworks for UIDT. To showcase the ability of the

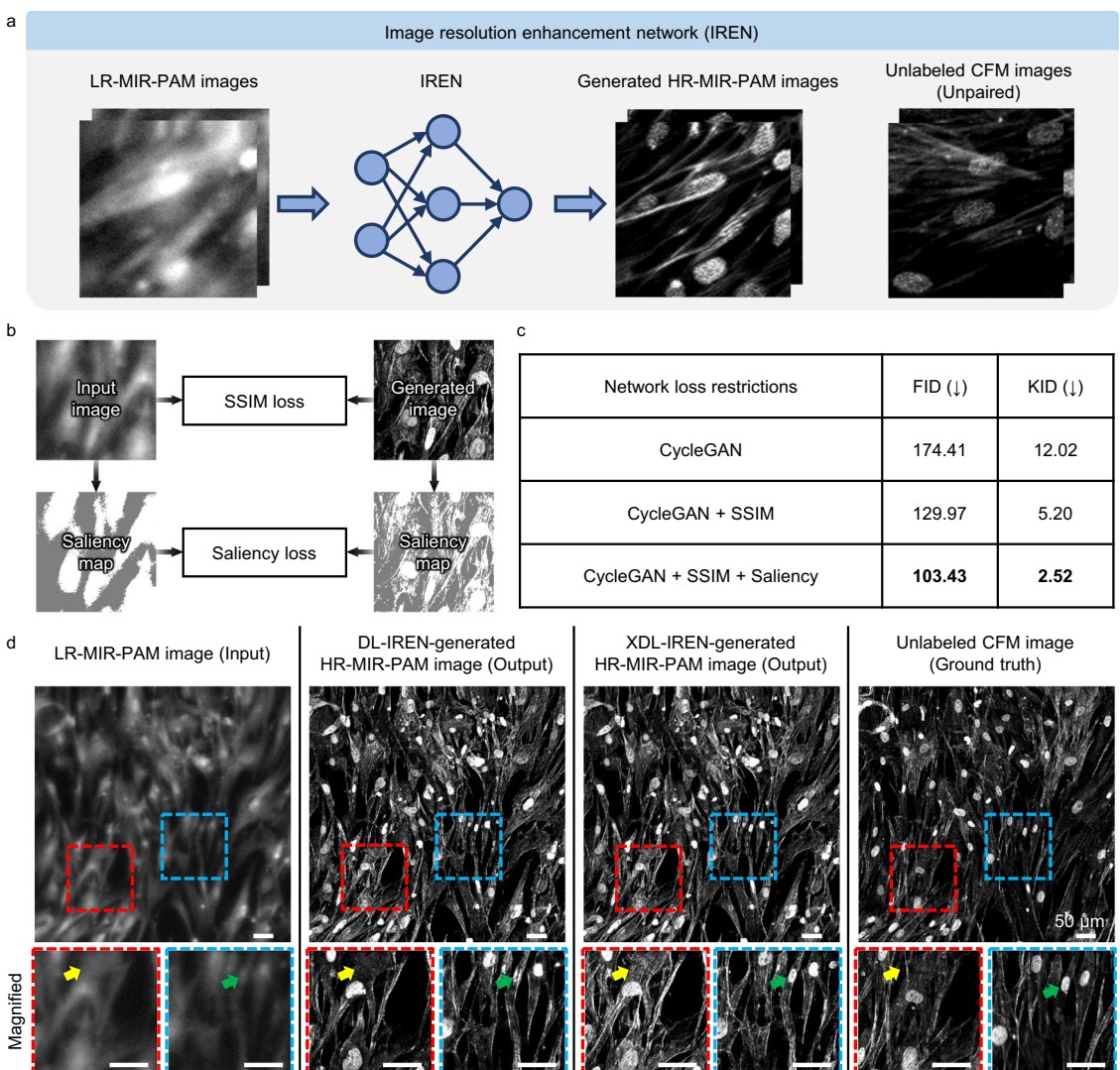

**Fig. 3 | XDL network for image resolution enhancement. a** Conceptual workflow of the image resolution enhancement network (IREN). **b** Loss restrictions for the XDL-IREN. The SSIM and saliency similarity refer to the preserved structural content between the image domains. **c** Quantitative comparison of IREN performance with network restrictions. Arrows in parentheses indicate the direction of better performance and the best scores are highlighted in bold font. **d** Visual comparison of IREN-generated images from the domains. In the HCF images with enhanced resolution, the oval and linear structures are inferred to be cell nuclei and F-actins, respectively. Scale bars, 50 μm. SSIM, structural similarity index; FID, Frechet inception distance; KID, kernel inception distance; DL, deep learning; and XDL, explainable deep learning.

UIDTs, we compared the performance of two framework configurations: (1) end-to-end and (2) pipeline. Both frameworks use the same LR-MIR-PAM images as input and translate them to virtually FL-stained HR-MIR-PAM ones. As shown in Fig. 5a, framework 1 (end-to-end) directly transforms to the target domain, whereas framework 2 (pipeline) predicts unlabeled HR-MIR-PAM images using the IREN and then predicts generated VS-HR-MIR-PAM ones using the VFSN. We also utilized saliency similarity in both frameworks to achieve explainability. The inset tables in Fig. 5a show the combinations of DL networks for comparing frameworks with explainability. For the DL and XDL, the CycleGAN and explainable CycleGAN are adopted, respectively. First of all, Fig. 5b compares the generated VS-HR-MIR-PAM images by using the XDL-UIDT frameworks. Framework 1 (Net 2) presents noticeably degraded sharpness and contrast in its VS images. Further, it also shows hallucinations of nuclei that cannot be seen in the ground truth images (yellow arrows). In contrast, Framework 2 (Net 6) successfully avoids these DL hallucinations and artifacts and produces sharp VS images, with a good match to the ground truth. We have further compared the performance of the frameworks with (XDL)

and without (DL) using saliency constraints in each network. When the images are transformed across two domains, content distortions can be prevented by applying the saliency constraint (Supplementary Fig. 6). The explainable frameworks can learn correct transformations for both resolution enhancement and virtual FL staining applications, and they achieve the best performance. By calculating the DL performance scores of the generated VS-HR-MIR-PAM images and corresponding CFM ones, we compared with different frameworks with or without explainable frameworks (Fig. 5c). The explainable framework 2, where saliency constraints are added in both IREN and VFLN (Net 6), significantly reduces the FID and KID values compared to the other networks, demonstrating it performs robustly with all the input image types studied.

We examined the biological features in HCF images to verify the feasibility of XDL-MIR-PAM. The properties are extracted after splitting channels from all tile images (Supplementary Fig. 7). From the blue channel, which mimics Hoechst staining of cell nuclei, we estimated the number, size, and aspect ratio of the cell nuclei. In addition, because the green channel mimics FITC staining for F-actin, we used it

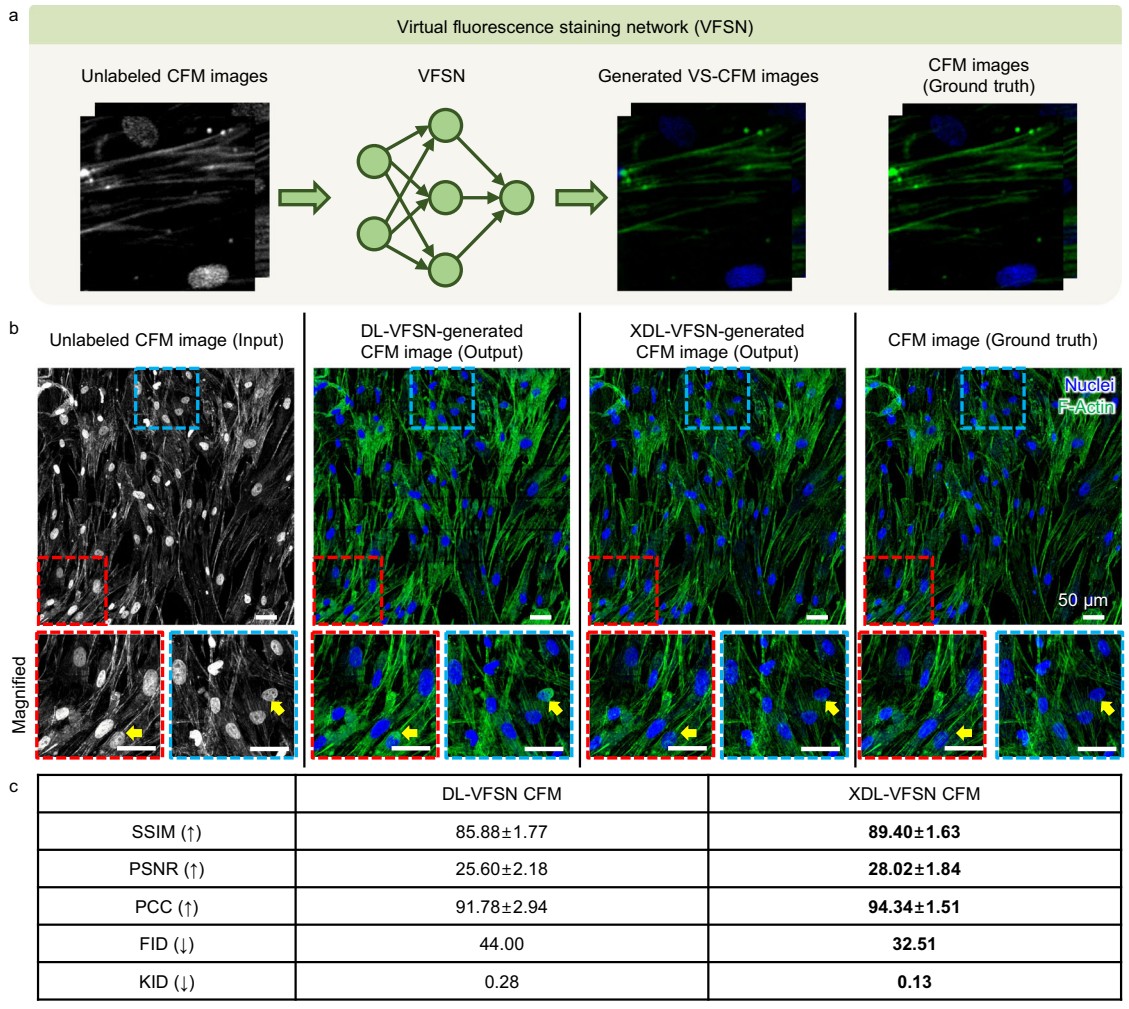

**Fig. 4 | XDL network for virtual FL staining. a** Conceptual workflow of the virtual fluorescence staining network (VFSN). **b** Visual comparison of VFSN-generated images between the domains. In the FL-stained HCF images, the blue and green channels refer to the staining by Hoechst and FITC, respectively. The yellow arrows indicate non-specific staining errors in the DL-VFSN but are corrected in the XDL-VFSN. Scale bars, 50 μm. **c** Table for quantitative comparison of VFSN performance. SSIM, structural similarity index; PSNR, peak signal-to-noise ratio; PCC, Pearson's correlation coefficient; FID, Frechet inception distance; and KID, kernel inception distance. Data are presented as mean ± SD. Arrows in parentheses indicate the direction of better performance and the best scores are highlighted in bold font. Source data are provided as a Source Data file.

to estimate the fibroblast area. The images generated by the XDL-MIR-PAM (Net 6) are quantitatively similar to the CFM images (Fig. 5d). Notably, in the fibroblast area, slightly higher result values of XDL-MIR-PAM imply that more structures were visualized with higher contrast compared to optically sectioned CFM images because the MIR-PAM has a greater depth-of-field (DOF) (Supplementary Fig. 8).

## Discussion

In this study, we introduced the XDL-UIDT, which implements image transformation to achieve VS-HR-MIR-PAM images. MIR-PAM is a promising imaging technology for identifying intrinsic properties of molecular bonds based on optical absorption without any chemical staining, but the spatial resolution for subcellular imaging is unsatisfactory due to the diffraction limitation of the long wavelengths. To overcome this, we devised the UIDT to transform the LR-MIR-PAM images into CFM-like ones. Using CycleGAN, an adequate network for unsupervised training, enables transforming the source domain into a target domain without the supervision of paired training data and time-consuming image registration. We first demonstrated label-free HCF imaging with protein selectivity using standalone MIR-PAM, then applied the UIDT for additive image processing with CFM. The UIDT, which consists of a two-step pipeline framework, overcame the limitations of conventional MIR-PAM by virtually improving resolution and FL staining. Furthermore, the proposed explainable framework ensured a content-preserving transformation by maintaining a similar saliency mask. Saliency loss helps address errors such as the misalignment of key features between the domains and the risk of focusing on irrelevant areas (e.g., background noise and artifacts) during the transformation. By monitoring the attention patterns, we can ensure that the model maintains attention on critical regions, and identify potential problems where the model focuses incorrectly during training. If the attention is on insignificant regions from the beginning, corrections can be made after a few iterations to shift the model's focus toward more relevant features. This insight enables us to adjust hyperparameters, retrain the model, and refine the dataset to improve performance and generalizability. These steps enhance the accuracy and interpretability of the UIDT. This enhancement can successfully avoid distortion of the image content and disorganization of semantic information, considerably improving stability and reliability and removing barriers to biological analysis. In addition, incorporating the physics of image formation into either the forward or backward mapping of CycleGAN reduces the number of network parameters and, more importantly, improves the quality of the transformation. The proposed XDL-based framework acts as an add-on

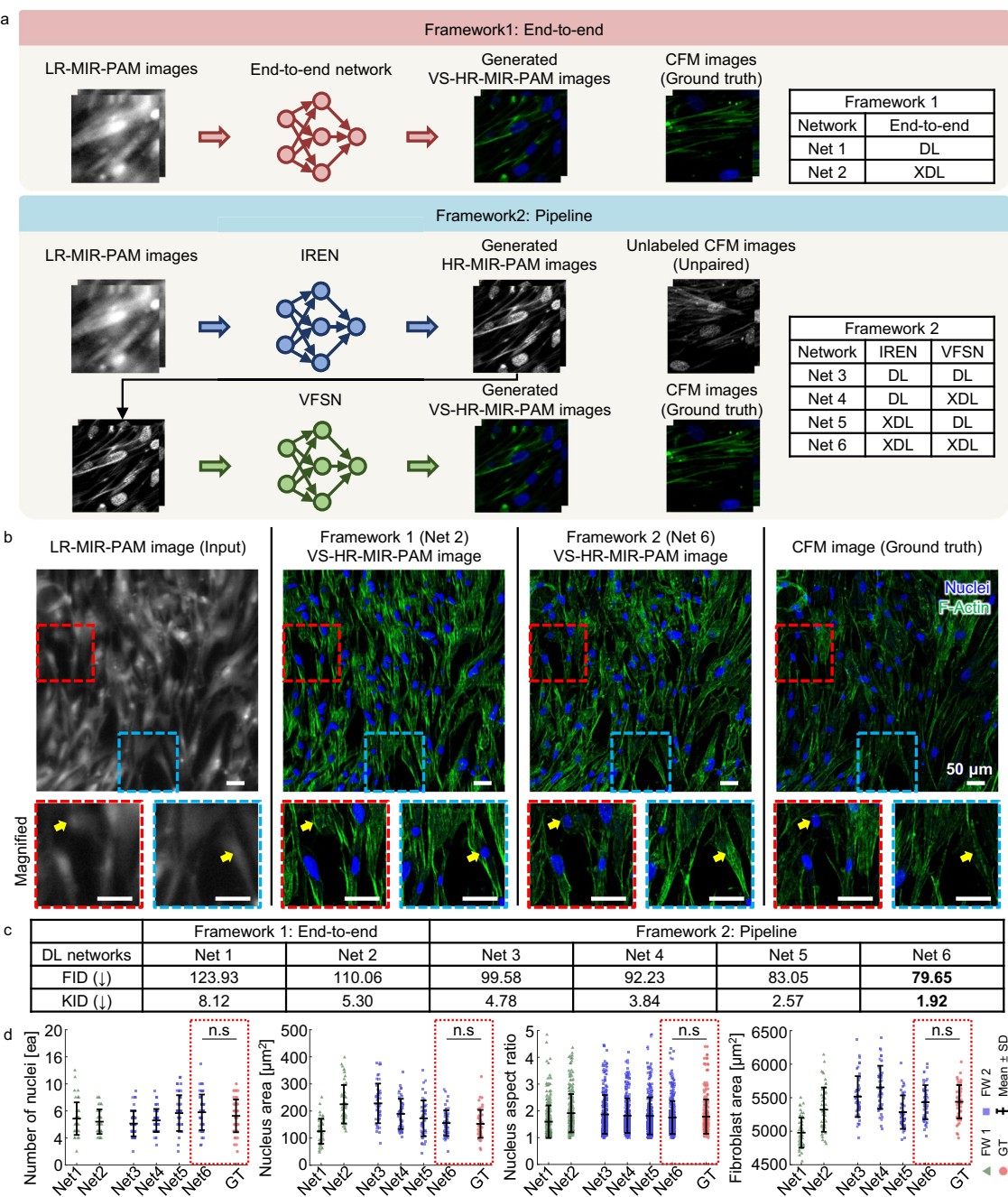

**Fig. 5 | Comparative performance of frameworks applied to the XDL-MIR-PAM.**
**a** Schematic of two types of frameworks. **b** Visual and (**c**)table for quantitative comparison between the networks. Scale bars, 50 μm. In the VS-HR-MIR-PAM images of HCFs, the blue and green channels refer to the cell nuclei and F-actins, respectively. The best scores are highlighted in bold font. **d** Quantitative comparisons of the number of nuclei, nucleus area, nucleus aspect ratio, and fibroblast area among various frameworks ($n = 49$, mean ± SD). For the DL and XDL, the CycleGAN and explainable CycleGAN are adopted, respectively. Significance by one-way ANOVA with Dunnett's multiple comparisons test: n.s, not significant ($p > 0.05$). $p = 0.7059$ (number of nuclei), $p = 0.9999$ (nucleus area), $p = 0.9564$ (nucleus aspect ratio), and $p = 1.0000$ (fibroblast area). Source data and $p$-values are provided as a Source Data file.

module to secure the robustness of the framework, producing more consistent images. The quality of transformed images was evaluated by comparing them to corresponding ground truth images. Quantitative results showed our proposed method can learn precise domain mappings and achieve state-of-the-art performance. To sum up, the XDL-MIR-PAM synergizes two imaging modalities (MIR-PAM and CFM), enabling label-free HR duplexed imaging in HCFs.

In terms of generalization, we applied the prebuilt framework to the disease model (Supplementary Fig. 9). In fibrotic conditions, quiescent fibroblasts can be transited to myofibroblasts with the upregulation of type I collagen and alpha-smooth muscle[40]. The activation by the transforming growth factor-beta (TGF-β) induces cell proliferation[41]. We observed a significant increase in cell number (2.1-fold: 0.17 to 0.36 ea) and fibroblast area (4.8-fold: 270 to 1301 μm²) using XDL-MIR-PAM. Moreover, we also confirmed that the nucleus area increased (1.4-fold: 172.5 to 235.1 μm²). The results of XDL-MIR-PAM in disease models are qualitatively and quantitatively similar to the CFM ones, and these morphological changes are consistent with the previous studies[42]. In addition, the superiority of XDL-MIR-PAM was demonstrated by visualizing living cells (Supplementary Note 2 and Supplementary Fig. 10). VS-HR-MIR-PAM images of label-free living HCF were predicted and biologically analyzed following the cell

growth. However, the performance of XDL-UIDT in living cells is lower than in fixed cells due to the low contrast in the original MIR-PAM images. Not only does the intracellular distinction diminish, but so does the cell boundary[43]. Aquatic cell culture media generates non-negligible PA signals induced by the MIR wavelengths, and thus the image contrasts from the living cells are relatively poor. Accordingly, to achieve efficient live-cell XDL-MIR-PAM imaging, the imaging platform, including the coupling media, needs to be optimized. Furthermore, the XDL-based framework requires delicate pre-processing (e.g., normalization and noise filter) and sufficient data collection (Supplementary Fig. 11). The core idea behind data augmentation is to add more samples or information to the training dataset, increasing the invariance and robustness of DL models. Here, we used horizontal and vertical flipping, tile overlapping, and rotation as augmentation techniques. We employed four-fold cross-validation to solve the overfitting problem where three WSI images are employed as a training set and one WSI image is used as a validation set in each fold. The FID & KID scores for each fold are shown in Supplementary Table 1.

We envision further improvements to this work. First, we can complement XDL networks with alternative state-of-the-art models. Recently, StyleGAN has been introduced to unravel high-level attributes with latent factors[44]. As the UIDT aims to translate between two different imaging modalities, stochastic effects in the generated image domain can provide controllability to the DL networks, ensuring improved performance, particularly in the IREN, with content preservation by explainable saliency similarity. Second, by embracing more FL channels with functional (spectral) information as well as structural information, HR multiplexed imaging for more diverse components can be achieved. Here, we have demonstrated a two-channel VFSN to distinguish HCF's cell nuclei and F-actins. Co-culture with other cells or further understanding of detailed microenvironments would require additional contrast channels, and multi-channel VFSN would allow observation of subcellular compositions and dynamics without biochemical labeling[45]. Moreover, cells undergo morphological and physiological changes in pathological situations[46]. In various disease situations, comprehensive identification of abnormality can be achieved by further training with complemental information[47]. Understanding pathophysiological phenomena (e.g., cellular behaviors and morphological changes) through DL-assisted assessment can be helpful for practical biological research. Third, XDL-UIDT performance and accuracy will be further improved with additional steps. DL-based networks (e.g., cell segmentation and classification) can be integrated into the framework. Interconnected DL framework enhances feature recognition, which aids in addressing artifact issues[48]. Fourth, the improved MIR-PAM can achieve superb volumetric images. The CFM provides optically sectioned 2D images, whereas the MIR-PAM can reconstruct 3D images, resulting in differences between images. However, poor spatial resolution and a shallow DOF impede the reconstruction of HR volumetric images. A metasurface-assisted MIR-PAM can increase the DOF and improve the lateral resolution[49]. In addition, a broadband UST can enhance sensitivity and axial resolution[50]. Bridging the gap between inter-domain images will enhance the performance of the XDL-UIDT. In conclusion, the XDL-UIDT can be extended to provide stable transformation across various imaging modalities and labeling protocols. We believe that our XDL-MIR-PAM provides a new blueprint for cell biology research.

## Methods

### MIR-PAM system
We used a pulsed quantum cascade laser (QCL) (MIRcat, Daylight Solutions) as a light source for the MIR-PAM system. The QCL is tunable in the spectral range of 5.55–7.35 μm (1801–1360 cm⁻¹), covering the high optical absorption bands of proteins, and it has a linewidth of about 1 cm⁻¹ (full width of half maximum). The pulse repetition rate was set to 100 kHz, with a pulse width of 20 ns. The laser beam was expanded and collimated by using ZnSe plano-convex lenses (#11-419 and #11-421, Edmund Optics) and focused on target samples through a 36x reflective objective lens (50102-02, Newport). A 12.5 mm diameter ZnSe window (#68-511, Edmund Optics) was adopted as a sample plate and attached to a water bath filled with PA coupling media. Heavy water (Deuterium oxide, 151882, Sigma-Aldrich) and cell culture media were used for the fixed and living cell imaging, respectively. Generated PA waves were detected via a customized UST with a center frequency of 30 MHz, a focal length of 4.5 mm, and an aperture diameter of 6 mm. The raw PA signals were amplified and filtered by two low-noise amplifiers (ZFL-500LN +, Mini-Circuits) with a gain of about 56 dB and a low-pass filter (ZX75LP-40-S +, Mini-Circuits). The signals were captured by a data acquisition (DAQ) board (NI PCIe-6321, National Instruments) and a 12-bit digitizer (ATS9350, AlazarTech, sampling rate: 250 MS/s). The imaging system was synchronized via a DAQ program developed in LabVIEW software (LabVIEW 2017, National Instruments). Stage-raster scanning was driven by two motorized linear stages (L-406.10SD00, Physik Instrumente), and the laser fluence, detected by a mercury cadmium telluride amplified photodetector (PDAVJ10, Thorlabs), was used to calibrate the PA signal pixel-by-pixel.

### Cell preparation
HCFs (C-12375, Promocell) up to passage 6 were cultured in a Fibroblast Growth Medium 3 kit (C-23130, Promocell) supplemented with 1% penicillin-streptomycin (SV30010, Hyclone). The medium was changed every other day before use. For cell seeding, fibronectin (1:200 diluted, 356008, Corning) was diluted with 1X PBS (SH30028.02, Hyclone) and coated on ZnSe substrates. The cells were dissociated using TrypLE (12604-021, Thermo Fisher Scientific), resuspended with maintenance medium, and seeded on fibronectin-coated ZnSe plates. To activate cardiac fibroblasts into myofibroblasts, 20 ng/ml TGF-β1 (7754-BH-100, R&D systems) was treated to HCFs 24 h after cell seeding. The TGF-β1-containing medium was changed every other day.

### Immunofluorescence staining analysis
For immunofluorescence imaging, the ZnSe plates were washed with 1X PBS for 5 min and fixed with 4% paraformaldehyde (CBPF-9004, Chembio) for 10 min. The samples were then washed three times with 1X PBS for 5 min per wash. The samples were permeabilized with a 0.1% Triton X-100 (T1020, Biosesang) solution for 10 min. Afterward, the samples were blocked with 5% normal goat serum (50062Z, Thermo Fisher Scientific) for 1 h. The cells were stained with fluorescein phalloidin (1:40 diluted, F432, Thermo Fisher Scientific) and Hoechst (1:1000 diluted, H3570, Thermo Fisher Scientific). The samples were imaged using the NIS-Elements advanced research software on an FL confocal laser scanning microscope (Nikon Ti Eclipse; Nikon). Three excitation wavelengths of 405, 488, and 594 nm were used to construct the blue, green, and red channels, respectively. We obtained and merged images at a magnification of 20x, corresponding to a pixel resolution of 0.31 μm/pixel.

### DL model: network architecture, training, and validation
To discover the unsupervised mapping between two image domains, both the IREN and VFSN used the GANs architectures, which were made up of two deep neural networks: a generator and a discriminator. The generator and discriminator utilized the traditional CycleGAN model design, as shown in Supplementary Fig. 1. The generator network adopted the ResNet-based model[51], consisting of a downsampling path, a residual path, and an upsampling path. The generator's first three layers used downsampling and strided convolution to extract low-level abstract representations. The first convolutional layer in the downsampling process increased the image channel while maintaining the same size, but the next two layers reduced and doubled the image size, respectively. Unlike the original generator in CycleGAN, we used a 3 × 3 kernel for the first

convolutional layer to keep the detailed features of the image. After the first layer, the channel size was increased to 64 while the image size stayed the same. The first layer was followed by two Convolution-InstanceNorm-ReLU layers in the downsampling route. These convolution layers were followed by a typical pooling layer with a stride. After passing through the downsampling layers, the image size was reduced by 2, but the channel number was increased by 2. The downsampling layers were followed by a lengthy residual neural network with 9 residual blocks. Each residual block was passed while maintaining the same image size and channel number. High-level characteristics were extracted by residual blocks. The number of residual blocks indicated the model's capacity. It is worth noting that more residual blocks are recommended for more complex tasks. The upsampling blocks lowered the number of channels by using three convolution layers with activation functions after bilinearly scaling the tensors twice. After each layer of upsampling, the image size was doubled by two, while the channel number was cut in half. After two layers of upsampling, the image size was restored to its former size, and the number of channels was reduced to 64. The channel number of the image was restored to 3 with the aid of the final Convolution layer. Strided convolution was also used to implement the final three layers of the upsampling process. They were applied to the image to rescale it to its original size and combine the extracted features. Features at different scales can be learned by using skip connections between the downsampling and upsampling layers at the same level. Due to the wide variety of microscopy images, the two GANs must have identical input and output channel numbers. Padded convolutional layers were employed to ensure that the image size was maintained while passing through the convolutional layers. Five blocks in the discriminator, each made with two convolutional layers and leaky ReLU pairs, collectively doubled the number of channels. The next layer was a two-stride average pooling layer. Following the five blocks, two fully connected layers decreased the output dimensionality to a single number, which was then used to determine the likelihood that the input to the discriminator network was a 1-dimensional output (either real or fake). The input of the discriminator network was either the virtually stained images from the generator or the immunofluorescent stained ground truth images. The input image was divided into small tiles, and classification was carried out at the tile scale. The final result was the same as the mean classification loss across all tiles.

## Loss function

We start by gathering two image sets (A and B) to sample the source domain and the target domain. To learn a pair of opposing mappings between the two image domains, a forward GAN and a backward GAN are trained simultaneously. A forward GAN uses an image from domain A (designated as a), the generator $G_A$ creates the new output image (referred to as $G_A(a)$), and the corresponding discriminator $D_B$ determines whether $G_A(a)$ is false or true. In a backward GAN, to distinguish between $G_B(b)$ and the images in domain A $D_A$ is employed to build a new image $G_B(b)$ based on image b in domain B. The loss function for the proposed method is made up of the losses of the forward and backward GAN ($L_{GAN}$), the cycle consistency loss ($L_{CYC}$), the SSIM loss ($L_{SSIM}$), and the saliency loss ($L_S$). For a forward GAN and backward GAN, the loss function is expressed as

$$L_{GAN}(G_A, D_B) = E_b\left[\left(D_B(b) - 1\right)^2\right] + E_a\left[\left(D_B(G_A(a))\right)^2\right] \quad (1)$$

$$L_{GAN}(G_B, D_A) = E_a\left[\left(D_A(a) - 1\right)^2\right] + E_b\left[\left(D_A(G_B(b))\right)^2\right] \quad (2)$$

The expectation for the random variables in each domain is shown by the symbol $E[\ldots]$. The process for $G_B(G_A(a))$ is characterized as the forward GAN, whereas the procedure for $G_A(G_B(a))$ is the backward GAN. The loss values $||G_B(G_A(a)) - a||_1$ and $||G_A(G_B(b)) - b||_1$ should

be reduced if the model is properly trained. The cycle consistency loss is therefore described as

$$L_{CYC}(G_A, G_B) = E_a\left[||G_B(G_A(a)) - a||_1\right] + E_b\left[||G_A(G_B(b)) - b||_1\right] \quad (3)$$

The identity loss in the original CycleGAN model[33], which was intended to maintain the color of input images, was dropped because it was ineffective in maintaining the image content. The SSIM loss[52] which helps preserve structural similarity between real and cycle-reconstructed images is expressed as

$$L_{SSIM}(G_A, G_B) = E_a\left[1 - SSIM(a, G_B(G_A(a)))\right] + E_b\left[1 - SSIM(b, G_A(G_B(b)))\right] \quad (4)$$

To achieve content-preserving transformation, we additionally imposed a saliency loss function. This loss function[23] is based on the finding that the backgrounds of microscope images, unlike the backgrounds of natural scenes, have comparable intensities. The loss was created to keep consistent the content masks extracted by using threshold segmentation:

$$L_S(G_A, T_A, T_B) = E_a\left[||\text{sigmoid}((a - T_A) * 100) - \text{sigmoid}((G_A(a) - T_B) * 100)||_1\right] \quad (5)$$

$$L_S(G_B, T_B, T_A) = E_b\left[||\text{sigmoid}((b - T_B) * 100) - \text{sigmoid}((G_B(b) - T_A) * 100)||_1\right] \quad (6)$$

$$L_S(G_A, G_B, T_A, T_B) = L_S(G_A, T_A, T_B) + L_S(G_B, T_B, T_A) \quad (7)$$

Here, the segmentation operators $T_A$ and $T_B$ are parameterized by thresholds for domains A and B, respectively. To produce a satisfactory saliency mask, the ideal thresholds were manually determined through experimentation. Finally, the full loss function can be expressed as

$$L(G_A, G_B, D_A, D_B, T_A, T_B) = L_{GAN}(G_A, D_B) + L_{GAN}(G_B, D_A) + \lambda L_{CYC}(G_A, G_B) \\ + \xi L_{SSIM}(G_A, G_B) + \rho L_S(G_A, G_B, T_A, T_B) \quad (8)$$

where $\lambda$, $\xi$, and $\rho$ are weighting constants to impose the cycle-consistency loss, SSIM loss, and saliency loss, respectively.

## Quantitative image metric

We utilized the PSNR[53], PCC[54], SSIM[52], FID[55], and KID[56] as evaluation indicators to assess the accuracy of the network output images quantitatively. They are described as

$$PSNR(a, b) = 10 \times \frac{MAX_I^2}{MSE} \quad (9)$$

where $MAX_I$ is the maximum possible value of the ground truth image. The mean square error (MSE) is defined as

$$MSE = \frac{1}{n^2}\sum_{i=0}^{n-1}\sum_{j=0}^{n-1}[I(i,j) - K(i,j)]^2 \quad (10)$$

where $I$ stands for the target image, while $K$ represents the image compared to $I$.

$$PCC(a, b) = \frac{\sigma_{ab}}{\sigma_a \sigma_b} \quad (11)$$

$$SSIM(a, b) = \frac{(2\mu_a\mu_b + c_1)(2\sigma_{ab} + c_2)}{(\mu_a^2 + \mu_b^2 + c_1)(\sigma_a^2 + \sigma_b^2 + c_2)} \quad (12)$$

where the two images being compared are denoted by $a$ and $b$. The mean values of $a$ and $b$ are represented by $\mu_a$ and $\mu_b$, respectively. The standard deviations of $a$ and $b$ are represented by $\sigma_a$ and $\sigma_b$, respectively, and $\sigma_{ab}$ stands for the cross-covariance of $a$ and $b$. The constants $c_1$ and $c_2$ are used to prevent division by zero.

The FID and KID are used to evaluate the visual quality of generated images, and both measure the distribution divergence between the generated images and the real images[57]. These metrics are the most well-accepted for measuring the images generated by unsupervised image translation models. For each pair of compared image sets, the FID creates a Gaussian distribution from the hidden activations of InceptionNet and then calculates the Fréchet distance between those Gaussians. The Fréchet distance is used to evaluate the quality of generated images, where a lower FID indicates a smaller distance between the real and target images. KID is a measure comparable to the FID, but it uses the squared maximum mean discrepancy (MMD) between Inception representations with a polynomial kernel. KID, in contrast to FID, offers a straightforward, unbiased estimator, making it more trustworthy, particularly when there are many more inception feature channels than image numbers. A lower KID indicates a higher visual similarity between real and predicted images. Our implementation for FID and KID is based on https://github.com/toshas/torch-fidelity.

## Implementation details
The image pre- and post-processing steps were implemented in MATLAB (R2021b, The MathWorks Inc.). The neural networks were implemented using Python 3.10.12, CUDA 11.8.0, and PyTorch 2.1.0. The training was carried out using an Intel Core i7 CPU, 32 GB of RAM, and two Nvidia GeForce RTX 2060 SUPER graphics processors (GPUs).

## Statistics and reproducibility
At least three independent biological samples were collected for every experimental group. The investigators were not blinded to allocation during experiments and outcome assessment. However, in the training phase, data were randomly divided and shuffled for unsupervised learning. Representative experiments for the network module performance of IREN (Fig. 3) and VFSN (Fig. 4) were performed using 96 test tiles. No data were excluded from the analysis. Statistical analyses were performed using $t$ tests, one-way or two-way ANOVA, assuming equal variance. Sample sizes are indicated in the figure legends, and significance was defined as $p$-values $< 0.05$. Data are presented as mean $\pm$ SD (standard deviation). Statistical analyses were conducted using GraphPad Prism 10 software.

## Reporting summary
Further information on research design is available in the Nature Portfolio Reporting Summary linked to this article.

## Data availability
The XDL-MIR-PAM data generated in this study and test data have been deposited in the Zenodo database [https://doi.org/10.5281/zenodo.14062532][58]. The training datasets are available from the corresponding author upon request, for research purposes. The sample test code with pre-trained networks is provided in GitHub: https://github.com/YoonChiHo/XDL_MIR_PAM_2024[59]. Source data are provided in this paper.

## Code availability
The code is available at https://github.com/YoonChiHo/XDL_MIR_PAM_2024[59].

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

## Acknowledgements

This work was supported by the following sources: Basic Science Research Program through the National Research Foundation of Korea (NRF) funded by the Ministry of Education (2020R1A6A1A03047902 received by C.K.), NRF grant funded by the Ministry of Science and ICT (MSIT) (2023R1A2C3004880 received by C.K.; 2021M3C1C3097624 received by C.K.), Korea Medical Device Development Fund grant funded by the Korea government (MSIT, the Ministry of Trade, Industry and Energy, the Ministry of Health & Welfare, the Ministry of Food and Drug Safety) (Project Number: 1711195277, RS-2020-KD000008 received by C.K.; 1711196475, RS-2023-00243633 received by C.K.), Korean Fund for Regenerative Medicine funded by MSIT, and Ministry of Health and Welfare (21A0104L1), Korea Institute for Advancement of Technology (KIAT) grant funded by the Korea Government (MOTIE) (P0008763 received by J.A.; P0021109 received by J.J.), Institute of Information & communications Technology Planning & Evaluation (IITP) grant funded by the Korea government (MSIT) (No. RS-2019-II191906, Artificial Intelligence Graduate School Program (POSTECH) received by C.K.), BK21 FOUR program, and Glocal University 30 projects.

## Author contributions

E.P., D.K., and J.A. developed the MIR-PAM imaging system. D.G.H. prepared HCFs and conducted CFM experiments. E.P., S.M., and C.Y. designed and carried out the XDL-UIDT. C.K. and J.J. supervised the project. All authors discussed the results and contributed to the writing.

## Competing interests

J.A. and C.K. have financial interests in OPTICHO, which, however, did not support this work. All other authors declare no competing interests.
