## [Transparent Peer Review file · Nature Communications]

Unsupervised inter-domain transformation for virtually stained high-resolution mid-infrared photoacoustic microscopy using explainable deep learning

Corresponding Author: Professor Chulhong Kim

Version 0:

Reviewer comments:

Reviewer #1

(Remarks to the Author)

The manuscript introduces a new technique for high-resolution biochemical imaging that does not require labeling. Utilizing mid-infrared photoacoustic (MIR-PA) imaging, this approach achieves label-free imaging capabilities. A novel computational algorithm is employed to enhance spatial resolution, lending considerable novelty and technical robustness to the methodology. The experimental results are compelling, showcasing a significant improvement in resolution. This advancement represents a breakthrough in technology with the potential for wide-reaching implications in various biomedical applications.

Regarding the technical specifics, I have several observations and requests for clarification:

- (1) The statistical data presented in Figure 2 should include the sample size for clarity. Similarly, in Figure 4c, it is necessary to specify whether the uncertainties depicted are standard errors or variations, along with the associated sample sizes.
- (2) The manuscript currently provides the resolution of the MIR-PAM system. However, it would be beneficial to present quantitative results demonstrating the extent of resolution enhancement achieved by the algorithm.
- (3) I infer that the method may be more applicable to thin samples due to the substantial attenuation of mid-infrared light in thicker materials. It would be useful to define the maximum sample thickness for optimal imaging. Additionally, a discussion on how performance diminishes with increased sample thickness would be valuable.
- (4) As mentioned by the authors, the efficacy of most numerical methods is contingent upon the signal-to-noise ratio (SNR) of the initial data or images. Therefore, I recommend providing a quantitative analysis of how initial SNR levels impact the effectiveness of image enhancement.

(Remarks on code availability)

Reviewer #2

(Remarks to the Author)

This work presents a notable contribution in the field of MIR photoacoustic microscopy (MIR-PAM) by introducing an unsupervised, explainable deep-learning model termed XDL-MIR-PAM. This model not only enhances the resolution of MIR-PAM but also enables virtual staining of the images acquired on unstained tissue samples. This technology holds significant potential for various biological applications, particularly in virtual histology.

One major advantage of XDL-MIR-PAM is that it achieves high resolution without the need for UV light localization, thus

mitigating the concerns about potential photodamage. Furthermore, its hardware configuration is relatively simple. To validate the resolution improvement and virtual staining effect, its performance was compared against the gold standard of confocal fluorescence microscopy (CFM).

I only have a couple of concerns:

1. It seems that the XDL model needs to be re-trained with CFM data whenever a new protein contrast is involved. It would be helpful to understand the time-consuming nature of the training process and how this new training might impact the performance of the XDL model on virtual staining of proteins for which the model is already trained.

2. While XDL-MIR-PAM has demonstrated excellent performance in imaging and virtual staining of normal tissue samples, it would be important to know how it performs when applied to diseased samples. Can it effectively identify abnormalities in cellular structures? Does achieving this require additional training to enable abnormality identification?

(Remarks on code availability)

Reviewer #3

(Remarks to the Author)

My comments are attached in the PDF.

(Remarks on code availability)

Version 1:

Reviewer comments:

Reviewer #1

(Remarks to the Author)

The manuscript presents a deep learning method for enhancing the resolution of MIR-PAM using a CycleGAN framework. By leveraging label-free imaging capabilities, this approach can be applied to a wide range of biomedical applications. The conclusions are well-supported by high-quality results, and the paper is excellently presented.

A major advantage and novelty of this method is its use of only MIR light to achieve high-resolution, label-free imaging of various absorbers. Compared with the UV-MIR-PA method, this deep-learning method is much easier to implement and thus can be readily available for many labs. It represents a significant advancement in the field

The training and validation of the deep learning model appear to be correct. The results reveal some common deep learning issues, such as hallucinations and potential overfitting. Although not completely resolved, the authors have demonstrated methods to mitigate these problems. In my opinion, these issues do not compromise the novelty and significance of the method. Thus, I recommend accepting the paper in its current form.

(Remarks on code availability)

Reviewer #2

(Remarks to the Author)

The authors have addressed all my concerns. I recommend the publication of this work.

(Remarks on code availability)

Reviewer #3

(Remarks to the Author)

The authors have conducted additional experiments in different imaging settings (e.g., diseased samples) and enhanced the pre-processing and data augmentation pipeline to show the reliable performance and generalizability of the XDL-UIDT. The authors claim that incorporating saliency loss and utilizing the GradCAM technique enhances the explainability of their deep learning model. The manuscript would benefit from additional discussion of how these methods contribute to the model's interpretability. While the performance improvement using saliency loss is evident, the manuscript would benefit from a more detailed exploration of the explainability aspect. Please address these points in Discussion.

Specific comments:

Saliency Maps The authors state that "Saliency loss continuously tracks saliency masks for both image domains to address unexpected errors that inevitably occur during the training process (Supplementary Fig. 2)." However, they do not elaborate

on the nature of these unexpected errors or how saliency loss specifically mitigates them. For example: What types of errors are being addressed? How does tracking saliency masks help in reducing these errors? How can researchers or clinicians know there is an unexpected error by tracking the saliency masks? Do saliency masks help uncover network biases or a consistent focus on irrelevant artifacts or structures?

GradCAM Heatmaps Similarly, the authors mention using GradCAM to "further explain the inner behavior of the model during the domain transformation from LR to HR (Supplementary Fig. 3)." However, the manuscript does not provide sufficient details on what specific information the GradCAM heatmaps reveal. For example:

Are the heatmaps highlighting key features like edges, textures, or anatomical structures that contribute to resolution enhancement? Are certain cellular features consistently highlighted, indicating where the model focuses during the transformation?

(Remarks on code availability)

Version 2:

Reviewer comments:

Reviewer #3

(Remarks to the Author)

All of my comments were addressed. I don't have further questions.

(Remarks on code availability)

REVIEWER COMMENTS

Reviewer #1

The manuscript introduces a new technique for high-resolution biochemical imaging that does not require labeling. Utilizing mid-infrared photoacoustic (MIR-PA) imaging, this approach achieves label-free imaging capabilities. A novel computational algorithm is employed to enhance spatial resolution, lending considerable novelty and technical robustness to the methodology. The experimental results are compelling, showcasing a significant improvement in resolution. This advancement represents a breakthrough in technology with the potential for wide-reaching implications in various biomedical applications.

Regarding the technical specifics, I have several observations and requests for clarification:

Reply: Thank you for your positive and constructive feedback on our manuscript.

Comment 1: *The statistical data presented in Figure 2 should include the sample size for clarity. Similarly, in Figure 4c, it is necessary to specify whether the uncertainties depicted are standard errors or variations, along with the associated sample sizes.*

Reply: We used 16 images each on day 1 and day 7 for label-free MIR-PAM imaging and the following quantitative analysis (Fig. 2). In addition, the quantification results presented in Fig. 4c indicated mean values and standard deviations in 98 test image tiles. We have revised the text on the main manuscript:

[Line 104] 16 images were used each on days 1 and 7.

[Line 168] We further quantified the performances of the VFSN by calculating the SSIM, peak signal-to-noise ratio (PSNR), Pearson's correlation coefficient (PCC), FID, and KID for 98 test image tiles (Fig. 4c).

[Line 178] Mean (\pm standard deviation) values are presented.

Comment 2: *The manuscript currently provides the resolution of the MIR-PAM system. However, it would be beneficial to present quantitative results demonstrating the extent of resolution enhancement achieved by the algorithm.*

Reply: The IREN generates HR-MIR-PAM images mimicking CFM ones. For quantitative evaluation of the XDL-IREN, we compared line profiles of each corresponding image. The XDL-IREN-generated HR-MIR-PAM images capture detailed structures of HCF (1–2 μm) beyond the resolution of LR-MIR-PAM (6–7 μm). We have added the related text in the Results section and Supplementary Fig.4.

[Line 146] Notably, the XDL-IREN-generated HR-MIR-PAM images can capture detailed structures of HCF (1–2 μm) beyond the resolution of LR-MIR-PAM (6–7 μm) (Supplementary Fig. 4).

Supplementary Fig. 4 | Verification for the HR-MIR-PAM.

Comment 3: I infer that the method may be more applicable to thin samples due to the substantial attenuation of mid-infrared light in thicker materials. It would be useful to define the maximum sample thickness for optimal imaging. Additionally, a discussion on how performance diminishes with increased sample thickness would be valuable.

Reply: We agree that MIR-PAM is more applicable to thin samples such as cells due to the high optical attenuation of MIR light. We measured imaging depth performance with a sloped surgical suture. Reaching a depth of about 60.7 μm , the PA signal amplitude decreased by 6 dB. The laser power irradiated to the sample was 0.2 mW at a wavelength of 6.0 μm . We have added the related text to the Results section and Supplementary Note 1.

[Line 110] The developed MIR-PAM has a lateral resolution of about 6.6 μm with an imaging depth of about 60.7 μm (Supplementary Note 1 and Supplementary Fig. 1).

Supplementary Fig. 1 | Imaging performance of MIR-PAM system. e, PA MAP images of a sloped surgical suture. **f,** Normalized PA signal amplitude along the central depth of the suture.

Comment 4: As mentioned by the authors, the efficacy of most numerical methods is contingent upon the signal-to-noise ratio (SNR) of the initial data or images. Therefore, I recommend providing a quantitative analysis of how initial SNR levels impact the effectiveness of image enhancement.

Reply: Consistent preprocessing of input images is essential to ensure the stable performance of XDL-UIDT. Noise levels within an image determine image quality (e.g., SNR and CNR) and information recognition accuracy, which affects the performance and efficiency of deep learning networks. To assess the effect of image quality in the XDL-UIDT, we performed a test by adding Gaussian noise to the original MIR-PAM image and applying it to the prebuilt XDL framework (Net 6). As the noise variance increases in input images, foreground/background recognition decreases, which causes problems in deep learning output images. Furthermore, we quantitatively confirmed that the deep learning performance degrades based on the FID and KID scores, which are significantly increased with respect to noise. We have added the text in the Discussion section and Supplementary Fig. 9.

[Line 242] Furthermore, the XDL-based framework requires delicate pre-processing (e.g., normalization and noise filter) and sufficient data collection (Supplementary Fig. 9).

Supplementary Fig. 9 | XDL-UIDT performance according to the noise variances.

Reviewer #2

This work presents a notable contribution in the field of MIR photoacoustic microscopy (MIR-PAM) by introducing an unsupervised, explainable deep-learning model termed XDL-MIR-PAM. This model not only enhances the resolution of MIR-PAM but also enables virtual staining of the images acquired on unstained tissue samples. This technology holds significant potential for various biological applications, particularly in virtual histology.

One major advantage of XDL-MIR-PAM is that it achieves high resolution without the need for UV light localization, thus mitigating the concerns about potential photodamage. Furthermore, its hardware configuration is relatively simple. To validate the resolution improvement and virtual staining effect, its performance was compared against the gold standard of confocal fluorescence microscopy (CFM).

I only have a couple of concerns:

Reply: Thank you for your positive and helpful feedback on our manuscript.

Comment 1: *It seems that the XDL model needs to be re-trained with CFM data whenever a new protein contrast is involved. It would be helpful to understand the time-consuming nature of the training process and how this new training might impact the performance of the XDL model on virtual staining of proteins for which the model is already trained.*

Reply: We agree that whenever new protein contrasts are added the DL model should be retrained. The presented results have been trained using HCF images, which comprise two channels: cell nucleus and F-actin. However, several proteins (e.g. collagen and vimentin) are included in subcellular biomarkers, along with lipid droplets and mitochondria. To generate multiplexed image with various contrasts, multi-channel CFM images need to be trained. Furthermore, supplemental label-free MIR-PAM images with functional (spectral) information as well as structural information may be required. Additional data for channel expansion and learning robustness are essential, which increase training time. However, because MIR-PAM is capable of high-selectivity cellular imaging, the proposed XDL framework will hold excellent performances, particularly in virtual fluorescence staining. We have revised the text in the discussion:

[Line 252] Second, by embracing more FL channels with functional (spectral) information as well as structural information, HR multiplexed imaging for more diverse components can be achieved. Here, we have demonstrated a two-channel VFSN to distinguish HCF's cell nuclei and F-actins. Co-culture with other cells or further understanding of detailed microenvironments would require additional contrast channels, and multi-channel VFSN would allow observation of subcellular compositions and dynamics without biochemical labeling⁴⁴.

Comment 2: *While XDL-MIR-PAM has demonstrated excellent performance in imaging and virtual staining of normal tissue samples, it would be important to know how it performs when applied to diseased samples. Can it effectively identify abnormalities in cellular structures? Does achieving this require additional training to enable abnormality identification?*

Reply: Under several pathological circumstances, including fibrosis and cancer, cells undergo morphological changes such as enlargement, stretching, and epithelial to mesenchymal transition. The fibroblasts in this study show significant morphological differences from the normal state after TGF- β treatment, which is a key driver of fibrosis. In fibrotic conditions, TGF- β promotes fibroblast proliferation, leading to their transition into myofibroblasts, morphologically enlarged and irregular (e.g., star or web-shaped), with the upregulation of type I collagen and α -SMA. We have applied the proposed framework to the disease model and successfully identified the abnormality. We confirmed that cells proliferated rapidly upon TGF- β treatment, and the cell body size increased due to the transformation of quiescent fibroblasts into myofibroblasts. We have added the results in Supplementary Fig. 7 and revised text in the Methods and Discussion:

Supplementary Fig. 7 | XDL-MIR-PAM imaging of fibrotic HCFs.

[Line 290] To activate cardiac fibroblasts into myofibroblasts, 20 ng/ml TGF- β 1 (7754-BH-100, R&D systems) was treated to HCFs 24 hours after cell seeding. The TGF- β 1-containing medium was changed every other day.

[Line 230] In terms of generalization, we applied the prebuilt framework to the disease model (Supplementary Fig. 7). In fibrotic conditions, quiescent fibroblasts can be transitioned to myofibroblasts with the upregulation of type I collagen and alpha-smooth muscle³⁹. The activation by the transforming growth factor-beta (TGF- β) induces cell proliferation⁴⁰. We observed a significant increase in cell number (2.1-fold: 0.17 to 0.36 ea) and fibroblast area (4.8-fold: 270 to 1301 μ m²) using XDL-MIR-PAM. Moreover, we also confirmed that the nucleus size increased (1.4-fold: 172.5 to 235.1 μ m²). The results of XDL-MIR-PAM in disease models are qualitatively and quantitatively similar to the CFM ones, and these morphological changes are consistent with the previous studies⁴¹.

[Line 256] Moreover, cells undergo morphological and physiological changes in pathological situations⁴⁵. In various disease situations, comprehensive identification of abnormality can be achieved by further training with complementary information⁴⁶. Understanding pathophysiological phenomena (e.g., cellular behaviors and morphological changes) through DL-assisted assessment can be helpful for practical biological research.

Reviewer #3

This manuscript presents a deep learning-based method for transforming low-resolution MIR-PAM images into high resolution while providing virtual staining. Confocal fluorescence microscopy was used as ground truth for both training and validation. A CycleGAN framework was adopted as the main network architecture. Overall, the paper demonstrates clarity in presentation and well-structured content. However, the major claim regarding the utilization of explainable deep learning lacks robust support within the proposed methodology. The saliency loss was used during training process as an additional constraint to facilitate network performance, which diverges from conventional XAI/XDL practices. In addition, the proposed method suffered from potential overfitting and clear hallucinations. Critically, the manuscript lacks assessment regarding the generalizability of the proposed DL network, representing a significant gap in its validation process.

Reply: Thank you for your important and valuable comments on our manuscript.

Specific comments:

Comment 1: *It is questionable whether the presented paper actually used the concept of explainable AI. The main difference between the explainable network (XDL) and the regular network (DL) is the addition of the saliency loss. Saliency maps in explainable AI are typically used post hoc training to explain which parts of the input were most influential to a neural network's decision. This includes backpropagation-based or perturbations-based methods to visualize features (i.e. using saliency maps or heatmaps) relevant to the network's prediction. Unlike what was being done in this paper, saliency maps are not used during the training process but are applied to the trained models to interpret their prediction. In contrast, the approach in the paper integrates the saliency concept into the loss function itself. The added saliency loss is based on the difference between the input image and the transformed image, after applying a segmentation operator parameterized by thresholds. This is simply an added consistency loss to ensure that these masks are consistent across both domains to preserve image content. This is not how saliency is used as an explanation tool for AI, which typically does not directly influence model training but rather aims to provide interpretability after the model has been trained. Therefore, the major claim of using "explainable deep learning" is not substantiated by the proposed method in this paper.*

Reply: Sorry for the confusion. In addition to interpreting the predictions by adopting explainability via saliency masks in the training model, we also sought to improve the performance of UIDT, a transformation between different imaging modalities, by adding them directly to the loss function.

$$SL(G_A) = L1 \left(\left(\text{sigmoid}((a - T_A) * 100) \right), \left(\text{sigmoid}((G_A(a) - T_B) * 100) \right) \right), \quad (4)$$

$$SL(G_B) = L1 \left(\left(\text{sigmoid}((b - T_B) * 100) \right), \left(\text{sigmoid}((G_B(b) - T_A) * 100) \right) \right), \quad (5)$$

$$L_{SL}(G_A, G_B) = SL(G_A) + SL(G_B). \quad (6)$$

We checked the saliency masks according to the training epochs (Supplementary Fig. 2). Saliency loss continuously tracks saliency masks for both image domains to address unexpected errors that inevitably occur during the deep learning training process. In both networks, the saliency mask visualizes the consistency of the overall structure, which should always be maintained regardless of quality differences or staining.

Additionally, we included Grad-CAM heatmaps that highlight the class-affecting features to enhance explainability (Supplementary Fig. 3). It explains the behavior in each layer of the transformer in the XDL-based generator. Here, we can see that as the layer gets deeper, the blurred attention of the model is gradually transformed into attention that is sharp and structure-sensitive enough to generate a high-resolution image.

We have added these lines in the revised manuscript:

[Line 127] Saliency loss continuously tracks saliency masks for both image domains to address unexpected errors that inevitably occur during the training process (Supplementary Fig. 2). In addition, GradCAM³⁸ technique is used to further explain the inner behavior of the model during the domain transformation from LR to HR (Supplementary Fig. 3). GradCAM heatmaps show the model's attention in each transformer layer of the XDL-based generator.

Supplementary Fig. 2 | XDL-generated images and salience masks in a. IREN and b. VFSN according to the epochs.

Supplementary Fig. 3 | GradCAM heat map in each transformer layer of the XDL-based generator.

Comment 2: *Some sort of variations and perturbations to the training model need to be considered and included in the study. For example, using different samples with different cellular structures and different imaging settings. This is especially important for the inference stage to test generalizability of the proposed DL network. All testing datasets used in this study are very similar to the training data, which may be prone to overfitting and does not represent real world applications.*

Reply: Thanks for your comment. We have applied data augmentation in the training process to prevent overfitting in deep learning. One common method for enhancing a model's performance is data augmentation. It consists of a collection of techniques designed to artificially increase the quantity and variations of data samples included in the dataset. This is carried out because deep learning models perform best when there are more data samples available for training. To produce a more complete set of training examples, these transformations make specific changes to the original data. The main goal is to replicate the range of data that the model will see in real life, which will enhance its generalization capacity. Additionally, we have applied a 4-fold cross-validation technique where three WSI images were employed as the training set and one WSI image was used as the validation and test set in each fold. So, we hope that data augmentation and cross-validation techniques enhance the models' robustness and generalization ability.

As a result of generalization, we conducted additional experiments on different image settings (image contrast and SNR) and samples (disease model and living cell growth). Please refer to responses to Reviewer #1's comment #4, Reviewer #2's comment #2, and comment #4 below, respectively.

We have added these lines in the revised manuscript:

[Line 243] The core idea behind data augmentation is to add more samples or information to the training dataset, increasing the invariance and robustness of DL models. Here, we used horizontal and vertical flipping, tile overlapping, and rotation as augmentation techniques. We employed four-fold cross-validation to solve the overfitting problem where three WSI images are employed as a training set and one WSI image is used as a validation set in each fold. The FID & KID scores for each fold are shown in Supplementary Table 1.

Fold	1	2	3	4	Mean
FID (↓)	100.43	106.81	100.87	105.45	103.39
KID (↓)	1.73	2.28	1.64	2.21	1.96

Comment 3: *Fig. 3: the ground truth image is unpaired, it'd be nice to have a set of data with paired ground truth to show that XDL indeed provides better performance (e.g., less nucleus split, more detailed F-actin structures as pointed out by Fig. 3d).*

Reply: We completely agree with your viewpoint in this regard. When a paired dataset is used, the XDL network can perform better. We confirmed good performance using paired CFM images in the VSFN. However, in the IREN, exact paired datasets between different modalities cannot be obtained due to hardware limitations. Physically, MIR-PAM and CFM detect and visualize HCF with different sensitivity due to their unique characteristics (e.g. imaging contrast and DOF). Public data pairs do not even exist as the MIR-PAM system is newly developed. Additionally, in practice, after acquiring label-free MIR-PAM images with cultured cells, CFM images were obtained via a chemical process (i.e. permeabilization, washing, and immunostaining), which may slightly change the morphology. To overcome this uncertainty, we aim for unsupervised inter-domain transformation and employ the evaluating metrics of FID and KID scores that validate transformation methods using unpaired data [a–c].

[a] Chen, et al. "Reusing Discriminators for Encoding: Towards Unsupervised Image-to-Image Translation", DOI: 10.1109/CVPR42600.2020.00819

[b] Paavilainen, et al. "Bridging the gap between paired and unpaired medical image translation" *MICCAI Workshop on Deep Generative Models*. DOI: 10.1007/978-3-030-88210-5_4

[c] Kim, et al. "Learning to discover cross-domain relations with generative adversarial networks" *Proceedings of the 34th International Conference on Machine Learning*. PMLR 70:1857-1865, 2017.

Comment 4: *Supplemental Fig. 3: these LR-MIR-PAM images have much lower quality than those appeared in the main text and from these LR-MIR-PAM images, the nucleus and the F-actin are barely visible, yet the network generated detailed HR images with virtual staining. This result indicates overfitting. A more diverse training and testing datasets with different imaging conditions are necessary to validate the proposed method.*

Reply: Supplementary Fig 8 shows XDL-MIR-PAM images obtained using living cells. For the performance of XDL-UIDT, both fixed and living cell image sets are included in the training dataset. In particular, images of days 1 and 7 were trained, and images of days 1, 4, and 7 were tested. Although raw MIR-PAM images in living HCF have lower contrast than in fixed HCF, XDL-MIR-PAM has been demonstrated with delicate pre-processing and data augmentation. The detailed description is presented in the Discussion section and Supplementary Note 2.

[Line 236] In addition, the superiority of XDL-MIR-PAM was demonstrated by visualizing living cells (Supplementary Note 2 and Supplementary Fig. 8).

Supplementary Fig. 8 | XDL-MIR-PAM imaging of living HCFs.

Comment 5: Hallucination was still prominent in the best result presented (framework 2, net 6), see examples below following the white arrows and circles (captured from Fig. 5). Therefore, the claim that “Framework 2 (Net 6) successfully avoids these hallucinations and artifacts, and produces sharp VS images, with a good match to the ground truth” is not fully supported by the results presented. In addition, because of the hallucination, a later statement, “Notably, in the fibroblast area, a slightly higher result implies more structures were detected by the XDL-MIR-PAM than in the optically sectioned CFM images, because the MIR-PAM has a greater depth-of-field (DOF)”, is also not substantiated unless multiple CFM images acquired at different depths can be used to validate that the higher number of structures detected by XDL-MIR-PAM were indeed real.

Reply: Sorry for the confusing statements. In the main text, we have tuned the fluorescence intensity not to be saturated for all channels of CFM. When adjusting the brightness (1.3x) of the CFM image, the information (i.e. nucleus and F-actin) was revealed that was out of focus (Supplementary Fig. 6, marked with white arrows and circles). These elements are recognized in the original MIR-PAM images and also visualized in XDL-MIR-PAM. we can infer that XDL-UIDT performs well on MIR-PAM, which has a longer DOF than CFM.

In addition, by adopting XDL-UIDT, we reduced the hallucinations that appear in traditional DL-assisted virtual staining and presented the evidence through image comparison (Figs. 4b and 5b). We used label-free cultured cells, which may contain errors such as cell debris or contamination. LR-MIR-PAM detected condensed F-actin with high PA signal amplitudes, and it was virtually stained according to the probability of nuclei. On the other hand, pipelined XDL-UIDT improved sensitivity to identifying cell nuclei and F-actins. Compared to existing DL-based methods, it was implemented much closer to the ground truth. Unfortunately, the VS-HR-MIR-PAM (Net 6) image still has a few misinterpreted features (marked with asterisks). However, we expect to address these errors with additional steps. DL-based cell segmentation network can be integrated into the framework, which will enhance the performance and accuracy of XDL-UIDT.

According to the reply above, we have revised the text as follows:

[Line 203] Notably, in the fibroblast area, slightly higher result values of XDL-MIR-PAM imply that more structures were visualized with higher contrast compared to optically sectioned CFM images, because the MIR-PAM has a greater depth-of-field (DOF) (Supplementary Fig. 6).

[Line 259] Third, XDL-UIDT performance and accuracy will be further improved with additional steps. DL-based networks (e.g. cell segmentation and classification) can be integrated into the framework. Interconnected DL framework enhances feature recognition, which aids in addressing artifact issues⁴⁷.

Supplementary Fig. 6 | Magnified images of XDL-UIDT.

REVIEWER COMMENTS

Reviewer #1

The manuscript presents a deep learning method for enhancing the resolution of MIR-PAM using a CycleGAN framework. By leveraging label-free imaging capabilities, this approach can be applied to a wide range of biomedical applications. The conclusions are well-supported by high-quality results, and the paper is excellently presented.

A major advantage and novelty of this method is its use of only MIR light to achieve high-resolution, label-free imaging of various absorbers. Compared with the UV-MIR-PA method, this deep-learning method is much easier to implement and thus can be readily available for many labs. It represents a significant advancement in the field

The training and validation of the deep learning model appear to be correct. The results reveal some common deep learning issues, such as hallucinations and potential overfitting. Although not completely resolved, the authors have demonstrated methods to mitigate these problems. In my opinion, these issues do not compromise the novelty and significance of the method. Thus, I recommend accepting the paper in its current form.

Reply: We sincerely appreciate your constructive feedback on our manuscript. Your suggestions have greatly strengthened our work.

Reviewer #2

The authors have addressed all my concerns. I recommend the publication of this work.

Reply: We sincerely appreciate your helpful comments for improving our manuscript.

Reviewer #3

The authors have conducted additional experiments in different imaging settings (e.g., diseased samples) and enhanced the pre-processing and data augmentation pipeline to show the reliable performance and generalizability of the XDL-UIDT. The authors claim that incorporating saliency loss and utilizing the GradCAM technique enhances the explainability of their deep learning model. The manuscript would benefit from additional discussion of how these methods contribute to the model's interpretability. While the performance improvement using saliency loss is evident, the manuscript would benefit from a more detailed exploration of the explainability aspect. Please address these points in Discussion.

Reply: Thank you for your important and valuable feedback.

Specific comments:

Comment 1: *Saliency Maps The authors state that "Saliency loss continuously tracks saliency masks for both image domains to address unexpected errors that inevitably occur during the training process (Supplementary Fig. 2)." However, they do not elaborate on the nature of these unexpected errors or how saliency loss specifically mitigates them. For example: What types of errors are being addressed? How does tracking saliency masks help in reducing these errors? How can researchers or clinicians know there is an unexpected error by tracking the saliency masks? Do saliency masks help uncover network biases or a consistent focus on irrelevant artifacts or structures?*

Reply: Thank you for your helpful comments. The integration of saliency loss addresses key issues during the transformation, such as the misalignment of important features, the model's focus on irrelevant regions (e.g., noise or artifacts), and the amplification of unwanted details. By tracking saliency masks, we ensure the model consistently highlights relevant areas, preserving critical structures and preventing focus on irrelevant information. Incorporating saliency loss allows the transformation to preserve important information in input domain images and properly map them to the output counterparts. If the saliency masks show that the model focuses on irrelevant areas, such as background regions, noise, or artifacts, instead of the key features of interest, this indicates an error in the model's learning process, which can be corrected after a few iterations to improve focus. A sudden shift in the attention pattern, where important regions are ignored or attention is directed to non-informative areas, can signal that the model is not functioning as expected. Such patterns provide a clear indication that the model needs adjustment, retraining, or refinement in its attention mechanism to improve its focus and overall accuracy. Saliency masks provide a visual guide, revealing whether the model focuses on essential features or irrelevant artifacts. This continuous tracking and monitoring can result in higher-quality and more interpretable UDIT. We have revised the text in the Discussion:

[Line 224] Saliency loss helps address errors such as the misalignment of key features between the domains and the risk of focusing on irrelevant areas (e.g., background noise and artifacts) during the transformation. By monitoring the attention patterns, we can ensure that the model maintains attention on critical regions, and identify potential problems where the model focuses incorrectly during training. If the attention is on insignificant regions from the beginning, corrections can be made after a few iterations to shift the model's focus toward more relevant features. This insight enables us to adjust hyperparameters, retrain the model, and refine the dataset to improve performance and generalizability. These steps enhance the accuracy and interpretability of the UDIT.

Comment 2: *GradCAM Heatmaps Similarly, the authors mention using GradCAM to "further explain the inner behavior of the model during the domain transformation from LR to HR (Supplementary Fig. 3)." However, the manuscript does not provide sufficient details on what specific information the GradCAM heatmaps reveal. For example: Are the heatmaps highlighting key features like edges, textures, or anatomical structures that contribute to resolution enhancement? Are certain cellular features consistently highlighted, indicating where the model focuses during the transformation?*

Reply: Sorry for the insufficient statement. GradCAM heatmaps show the inner behavior during the training process. As shown in Supplementary Fig. 3, morphological and textural features in HCF that contribute to the transformation are visualized. As the layer progresses, the key feature becomes apparent. In particular, the cell nucleus is consistently highlighted and structurally distinguished from F-actin. We have added the text in the revised manuscript:

[Line 132] GradCAM captures the morphological and textural features in HCF that contribute to the transformation, and the key features become apparent as the layer progresses. Notably, the cell nucleus is consistently highlighted and structurally distinguished from F-actin.

This manuscript presents a deep learning-based method for transforming low-resolution MIR-PAM images into high resolution while providing virtual staining. Confocal fluorescence microscopy was used as ground truth for both training and validation. A CycleGAN framework was adopted as the main network architecture. Overall, the paper demonstrates clarity in presentation and well-structured content. However, the major claim regarding the utilization of explainable deep learning lacks robust support within the proposed methodology. The saliency loss was used during training process as an additional constraint to facilitate network performance, which diverges from conventional XAI/XDL practices. In addition, the proposed method suffered from potential overfitting and clear hallucinations. Critically, the manuscript lacks assessment regarding the generalizability of the proposed DL network, representing a significant gap in its validation process.

Specific comments:

1. It is questionable whether the presented paper actually used the concept of explainable AI. The main difference between the explainable network (XDL) and the regular network (DL) is the addition of the saliency loss. Saliency maps in explainable AI are typically used post hoc training to explain which parts of the input were most influential to a neural network's decision. This includes backpropagation-based or perturbations-based methods to visualize features (i.e. using saliency maps or heatmaps) relevant to the network's prediction. Unlike what was being done in this paper, saliency maps are not used during the training process but are applied to the trained models to interpret their prediction. In contrast, the approach in the paper integrates the saliency concept into the loss function itself. The added saliency loss is based on the difference between the input image and the transformed image, after applying a segmentation operator parameterized by thresholds. This is simply an added consistency loss to ensure that these masks are consistent across both domains to preserve image content. This is not how saliency is used as an explanation tool for AI, which typically does not directly influence model training but rather aims to provide interpretability after the model has been trained. Therefore, the major claim of using "explainable deep learning" is not substantiated by the proposed method in this paper.
2. Some sort of variations and perturbation to the training model needs to be considered and included in the study. For example, using different samples with different cellular structures and different imaging settings. This is especially important for the inference stage to test generalizability of the proposed DL network. All testing datasets used in this study are very similar to the training data, which may be prone to overfitting and does not represent real world applications.
3. Fig. 3: the ground truth image is unpaired, it'd be nice to have a set of data with paired ground truth to show that XDL indeed provides better performance (e.g., less nucleus split, more detailed F-actin structures as pointed out by Fig. 3d).
4. Supplemental Fig. 3: these LR-MIR-PAM images have much lower quality than those appeared in the main text and from these LR-MIR-PAM images, the nucleus and the F-actin are barely visible, yet the network generated detailed HR images with virtual staining. This result indicates overfitting. A more diverse training and testing datasets with different imaging conditions are necessary to validate the proposed method.

5. Hallucination was still prominent in the best result presented (framework 2, net 6), see examples below following the white arrows and circles (captured from Fig. 5). Therefore, the claim that “Framework 2 (Net 6) successfully avoids these hallucinations and artifacts, and produces sharp VS images, with a good match to the ground truth” is not fully supported by the results presented. In addition, because of the hallucination, a later statement, “Notably, in the fibroblast area, a slightly higher result implies more structures were detected by the XDL-MIR-PAM than in the optically sectioned CFM images, because the MIR-PAM has a greater depth-of-field (DOF)”, is also not substantiated unless multiple CFM images acquired at different depths can be used to validate that the higher number of structures detected by XDL-MIR-PAM were indeed real.